# XBIC: Shapley-Enhanced BIC for Accurate Causal Discovery in Discrete Bayesian Networks

## Abstract

Score-based causal discovery for purely discrete data remains dominated by hill-climbing with the Bayesian Information Criterion (BIC), yet BIC often struggles to orient edges within Markov-equivalence classes. We introduce XBIC, a principled enhancement that soft-weights BIC's complexity penalty with edge-specific Shapley evidence: when a candidate parent contributes strongly to its child's likelihood, XBIC reduces the penalty proportionally, while defaulting to standard BIC when support is weak. Across ten benchmark discrete Bayesian networks (6-76 nodes) and seven sample-size regimes (700 runs), XBIC yields consistent gains, improving oriented-edge $F_1$ by 5.6% over hill-climbing BIC, 9.6% over a generalized-score GES variant, and 20.9% over PC. XBIC remains a drop-in upgrade within the familiar BIC framework. To facilitate adoption and reproducibility, we release code, data splits, and scripts at https://anonymous.4open.science/r/causal_discovery_shap-6900/README.md.

## 1 Introduction

A causal relationship exists when changes in one variable systematically influence another. Bayesian networks (BNs) encode such relationships as a directed acyclic graph (DAG) $G$ over variables $X_1, \ldots, X_M$ together with local mechanisms $X_i := f_i(\text{PA}_i, U_i)$, where $\text{PA}_i$ are the parents of $X_i$ in $G$ and $U_i$ is exogenous noise Pearl (2011). This representation supports interventional reasoning by modifying one variable and assessing induced changes in others.

Causal structure discovery—the task of recovering $G$ from observational data—is challenging because it must separate causation from correlation, yet it is critical in domains such as medicine and public health Spirtes et al. (2000). We study score-based causal discovery from purely *discrete* data, a setting that has received relatively less attention than continuous cases Goudet et al. (2018); Shimizu et al. (2006); Bühlmann et al. (2014). Discrete data are ubiquitous (e.g., clinical codes, insurance categories, survey responses) Beinlich et al. (1989); Cooper & Herskovits (1992); Sachs et al. (2005) and present distinct challenges: nonlinear, non-Gaussian dependencies; high-cardinality conditional probability tables; and sparsity that weakens conditional-independence testing Tsamardinos et al. (2006).

Two method families dominate causal discovery. Constraint-based methods (e.g., PC Spirtes et al. (2000)) iteratively remove edges via conditional-independence tests and return a partially directed acyclic graph (PDAG) that encodes a Markov equivalence class. Score-based methods search over DAGs to optimize a decomposable criterion such as BIC or AIC Schwarz (1978); Akaike (1974). A shared limitation is difficulty resolving edge directions within Markov-equivalent graphs: colliders $(X_1 \to X_2 \leftarrow X_3)$ can be identified, but mediator and confounder motifs $(X_1 \to X_2 \to X_3$ vs. $X_1 \leftarrow X_2 \to X_3)$ both yield $X_1 \perp X_3 \mid X_2$, leaving orientations unresolved.

We address this limitation by augmenting the BIC score with directional evidence derived from local feature attributions. For each node $X_i$, we train a classifier to predict $X_i$ from $X_{\setminus i}$ and use TreeExplainer to compute Shapley values Lundberg & Lee (2017); Lundberg et al. (2018) that quantify the marginal contribution of $X_j$ to predicting $X_i$. Aggregated over samples, these attributions induce an edge-specific signal for $X_j \to X_i$. We integrate this signal by soft-weighting BIC's complexity penalty: edges with strong directional support are penalized less, whereas edges with weak or

ambiguous support retain the default penalty. The approach preserves the familiar BIC framework while injecting asymmetric, edge-level information that helps prefer a single DAG within an equivalence class. In contrast to prior work that uses causal knowledge to constrain explanations Frye et al. (2020) or designs causality-aware sampling for global importances Breuer et al. (2024), we use explanations to improve structure learning itself when the graph is unknown.

We evaluate on ten benchmark discrete BN structures (6-76 nodes) across seven sample-size regimes (700 runs). The proposed method (XBIC) consistently improves oriented-edge $F_1$ over hill-climbing BIC (by 5.6%), a generalized-score GES variant (by 9.6%), and PC (by 20.9%), while reverting to standard BIC where directional evidence is weak. These gains come with additional computational cost due to per-node classifiers and attribution aggregation; we report wall-clock comparisons and discuss parallelization.

Our key contributions are:

1. A score-based method (XBIC) for discrete causal discovery that integrates edge-specific Shapley evidence as a soft weight on BIC's complexity term, providing directional preference within Markov equivalence classes while defaulting to standard BIC when evidence is limited.

2. An extensive empirical study on ten discrete networks and seven sample-size regimes demonstrating consistent improvements in oriented-edge recovery against strong baselines, with released code, data splits, and scripts to ensure reproducibility.

## 2 RELATED WORK

### 2.1 CAUSAL GRAPH DISCOVERY FOR DISCRETE DATA

Discrete causal discovery is commonly approached with *constraint-based* and *score-based* methods. The PC algorithm Spirtes et al. (2000) identifies a graph skeleton and orients a subset of edges by performing conditional-independence (CI) tests among variable pairs $X_i$ and $X_j$ given subsets $S \subseteq X \setminus \{X_i, X_j\}$. The output is a completed partially directed acyclic graph (CPDAG/PDAG) that encodes a Markov equivalence class; edges whose direction cannot be determined remain undirected in the CPDAG.

Score-based approaches search over DAGs to optimize a decomposable criterion such as BIC or AIC Schwarz (1978); Akaike (1974). The BIC score balances fit and complexity,

$$\text{BIC}(G \mid D) = \log P(D \mid G) - \tfrac{\log N}{2} \dim(G), \tag{1}$$

with log-likelihood $\log P(D \mid G) = \sum_{d \in D} \log P(d \mid G)$. Although maximizing BIC over DAGs is NP-hard Cooper & Herskovits (1992), local heuristics such as hill climbing (with add/delete/reverse moves) Koller & Friedman (2009) and equivalence-class searches such as GES Chickering (2002) work well in practice. Hybrid algorithms (e.g., MMHC Tsamardinos et al. (2006)) first recover a skeleton via CI tests and then orient edges via a score-based search, often improving scalability on large sparse graphs.

A central limitation across these families is resolving directions within Markov-equivalence classes. While collider structures ($X_1 \rightarrow X_2 \leftarrow X_3$) are identifiable from CI patterns, mediator and confounder motifs ($X_1 \rightarrow X_2 \rightarrow X_3$ vs. $X_1 \leftarrow X_2 \rightarrow X_3$) both induce $X_1 \perp X_3 \mid X_2$. Our approach augments BIC with edge-specific directional evidence to prefer a single DAG when the CI pattern alone is insufficient, while reverting to standard BIC when such evidence is weak.

### 2.2 CAUSALITY AND EXPLAINABLE AI (XAI)

Most work at the causality–explainability interface injects causal knowledge to produce more faithful explanations, rather than using explanations to improve structure learning. Asymmetric Shapley values incorporate a known causal order into the coalition space for local explanations Frye et al. (2020). Causal Shapley values leverage (partially) known graphs to separate confounding from interaction in attributions Heskes et al. (2020). The choice between observational and interventional conditioning for "feature removal" has been analyzed as a key design degree of freedom for Shapley methods Janzing et al. (2020). Shapley Flow generalizes credit assignment from nodes to edges

on a *known* causal graph for global explanations Wang et al. (2021). These lines assume access to causal structure (full or partial) to constrain or reinterpret explanations. In contrast, we use local attributions—computed when the graph is unknown—to inform the scoring of candidate structures in a discrete, score-based pipeline.

## 2.3 SHAPLEY-GUIDED APPROACHES ADJACENT TO CAUSAL DISCOVERY

Recent efforts bring Shapley-inspired signals into pipelines adjacent to discovery. CAGE proposes a causality-aware sampling scheme for computing *global* Shapley importances Breuer et al. (2024). CD-RCA attaches Shapley scores to a time-series causal-diagnosis pass to localize root causes of forecast errors Yokoyama et al. (2024). ReX ranks edges with KernelSHAP prior to constraint pruning in *continuous* settings Renero et al. (2025). These methods assume real-valued features, explicit temporal order, or aim at explanation/diagnosis rather than discrete score-based structure learning.

To the best of our knowledge, our work is the first to directly integrate *local* feature attributions as an edge-specific, directional modulation of a *score-based* objective (BIC) for purely discrete data.

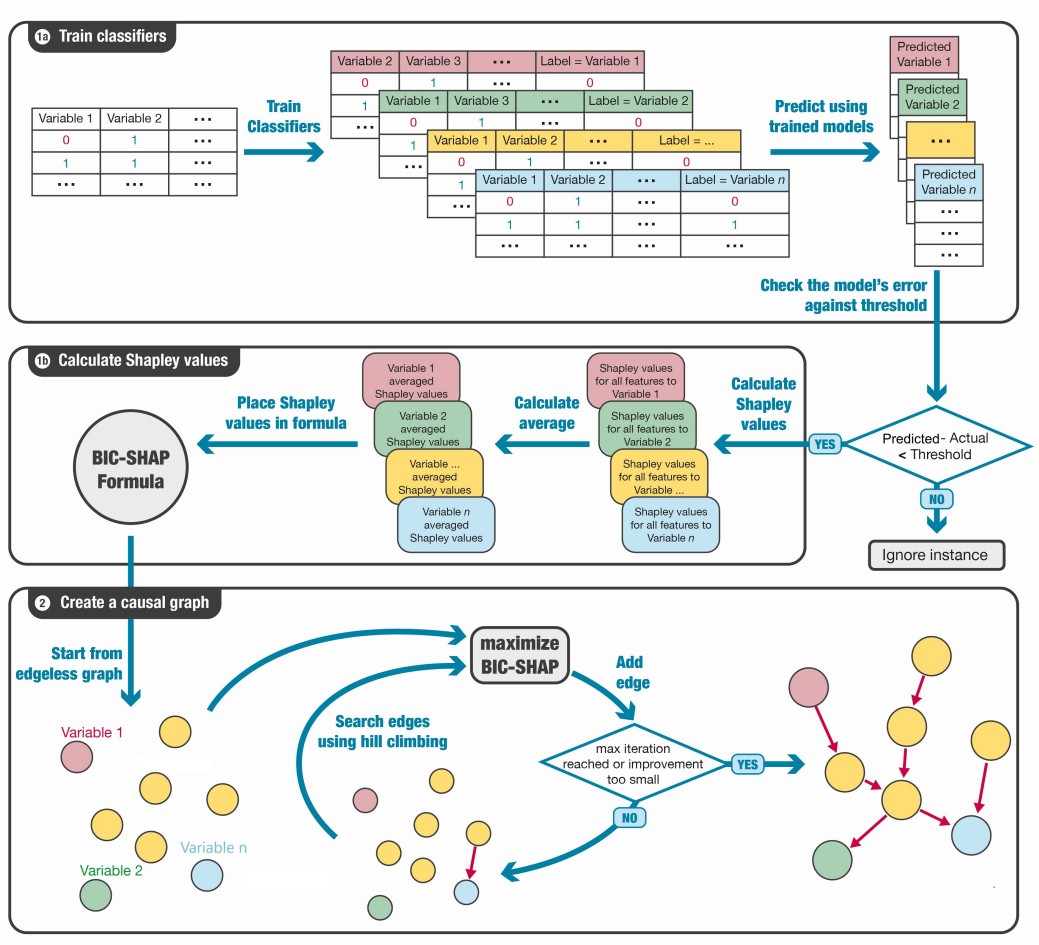

Figure 1: Overview of the XBIC pipeline. Stage 1: train a per-node classifier. Stage 2: compute and aggregate Shapley values. Stage 3: perform score-based search with a BIC term soft-weighted by directional evidence.

## 3 METHOD

---

**Algorithm 1** Mean directional attributions per target

---

**Require:** data matrix $D \in \mathbb{R}^{N \times M}$ (discrete columns), confidence threshold $\tau$
 1: **for** $i \leftarrow 1$ **to** $M$ **do**
 2:  $f_i \leftarrow \text{TRAINXGBOOST}(X_{\setminus i}, X_i)$
 3:  $S_i \leftarrow \{ n \in [N] : \max_c \Pr(X_i{=}c \mid X_{\setminus i}{=}x^{(n)}) \geq \tau \}$
 4:  Initialize accumulator $A_i \in \mathbb{R}^{M-1} \leftarrow 0$
 5:  **for** each $n \in S_i$ **do**
 6:   $\phi^{(n)} \leftarrow \text{TREEEXPLAINER}(f_i, x_{\setminus i}^{(n)})$   {exact SHAP for trees Lundberg et al. (2018)}
 7:   $A_i \leftarrow A_i + \phi^{(n)}$
 8:  **end for**
 9:  $\bar{\Phi}^{(i)} \leftarrow \begin{cases} A_i/|S_i|, & |S_i| > 0 \\ \mathbf{0}, & \text{otherwise} \end{cases}$
10:  For each $j \neq i$, set $\bar{\Phi}_{j \rightarrow i}$ to the $j$th component of $\bar{\Phi}^{(i)}$
11: **end for**
12: **return** $\{\bar{\Phi}_{j \rightarrow i}\}_{j \neq i,\, i=1}^M$

---

We propose *XBIC*, a score for structure learning on discrete data that augments the Bayesian Information Criterion (BIC) Schwarz (1978) with edge-specific, directional evidence derived from Shapley values Lundberg & Lee (2017). Shapley values are typically used to quantify a feature's contribution to a model's prediction; here we use them to induce an edge-level signal $X_j \rightarrow X_i$ and softly modulate BIC's complexity penalty during search.

Figure 1 summarizes the three stages.

1. **Stage 1: per-node predictors.** For each variable $X_i$, train a classifier $f_i : X_{\setminus i} \rightarrow X_i$.

2. **Stage 2: directional attributions.** For each $f_i$, compute Shapley values for inputs $X_j$ ($j \neq i$) and aggregate them over confidently predicted instances to obtain mean attributions $\bar{\Phi}_{j \rightarrow i}$.

3. **Stage 3: score-based search.** Use a hill-climbing search with an XBIC score that down-weights the BIC penalty in proportion to the aggregate directional evidence carried by the edges of the candidate graph.

**XBIC score.** Let $E(G)$ denote the edge set of DAG $G$ and $N$ the sample size. We define

$$\text{XBIC}_w(G \mid D) \;=\; \log P(D \mid G) \;-\; \frac{\log N}{2} \frac{\dim(G)}{\exp(w\,\text{SHAP}(G))}, \tag{2}$$

where $w \geq 0$ is a weight and

$$\text{SHAP}(G) \;=\; \sum_{(j \rightarrow i) \in E(G)} \big|\bar{\Phi}_{j \rightarrow i}\big|. \tag{3}$$

Thus, stronger aggregate evidence on the edges of $G$ (larger $\text{SHAP}(G)$) yields a smaller complexity penalty. Two immediate properties hold: (i) if $w = 0$ or $\text{SHAP}(G) = 0$, then $\text{XBIC}_w = \text{BIC}$; (ii) for any fixed $w$ and bounded $\text{SHAP}(G)$, the penalty still grows as $O(\log N)$, preserving BIC's order of penalization.

### 3.1 STAGE 1: PER-NODE CLASSIFIERS (FIGURE 1A)

For each target $X_i$ ($i = 1, \ldots, M$), we train a supervised model $f_i$ on inputs $X_{\setminus i}$ and obtain class probabilities $p_i(x) = \Pr(X_i{=}c \mid X_{\setminus i}{=}x)$. We instantiate $f_i$ with XGBoost Chen & Guestrin (2016) (five-fold CV for hyperparameters). For each sample $x^{(n)}$, let $\hat{c}^{(n)} = \arg\max_c p_i(x^{(n)})$ and $\kappa_i^{(n)} = \max_c p_i(x^{(n)})$ be the confidence. We retain the instance for attribution only if $\kappa_i^{(n)} \geq \tau$ for a fixed threshold $\tau \in (0, 1)$. This filter reduces attribution noise from low-certainty predictions and lowers the number of SHAP evaluations; sensitivity to $\tau$ is examined in Section 4.3.

### 3.2 STAGE 2: AGGREGATION (FIGURE 1B)

Given the confident index set $S_i$ for target $i$, the per-feature mean attribution is

$$\bar{\Phi}_{j \to i} \;=\; \frac{1}{|S_i|} \sum_{n \in S_i} \phi_j^{(n)}(f_i), \qquad j \neq i, \tag{4}$$

where $\phi_j^{(n)}(f_i)$ is the SHAP value of feature $X_j$ for model $f_i$ on instance $n$. Intuitively, if $|\bar{\Phi}_{1 \to 2}| \gg |\bar{\Phi}_{2 \to 1}|$, the edge $X_1 \to X_2$ has stronger directional support than $X_2 \to X_1$.

### 3.3 STAGE 3: HILL-CLIMBING WITH XBIC (FIGURE 1C)

We perform a standard local search over DAGs with add/delete/reverse moves, accepting the neighbor with the largest $\text{XBIC}_w$ improvement, subject to acyclicity. Caching local families keeps rescoring cost low.

---

**Algorithm 2** Hill-climbing with $\text{XBIC}_w$

---

**Require:** data $D$, weight $w$, attributions $\{\bar{\Phi}_{j \to i}\}$
 1: $G \leftarrow$ empty DAG                                                   (or a given prior)
 2: $s \leftarrow \text{XBIC}_w(G \mid D)$
 3: **repeat**
 4:    $s^\star \leftarrow s, \quad G^\star \leftarrow G$
 5:    **for** each valid local move $m \in \{\text{ADD}, \text{DEL}, \text{REV}\}$ producing $G'$ **do**
 6:       $s' \leftarrow \text{XBIC}_w(G' \mid D)$
 7:       **if** $s' > s^\star$ **then**
 8:          $s^\star \leftarrow s', \quad G^\star \leftarrow G'$
 9:       **end if**
10:    **end for**
11:    $G \leftarrow G^\star, \quad s \leftarrow s^\star$
12: **until** $G$ and $s$ unchanged
13: **return** $G$

---

**Computational considerations.** Let $M$ be the number of variables and $N$ the samples. (i) Training $M$ XGBoost models costs $O\big(M \cdot T(N, M)\big)$, where $T$ is the learner's training time. (ii) Exact TreeExplainer for trees Lundberg et al. (2018) is linear in the number of trees and their depth; computing and averaging SHAP on $|S_i|$ instances per target is parallelizable across $i$. (iii) Each search iteration evaluates $O(M^2)$ neighbors; with family caching, rescoring is local. Overall, XBIC adds a front-loaded attribution phase to a standard BIC search but parallelizes naturally across targets and (optionally) across instances.

**Consistency remark.** Write the penalty term in equation 2 as

$$\underbrace{\frac{\log N}{2}}_{\text{BIC growth}} \times \underbrace{\frac{\dim(G)}{\exp(w\,\text{SHAP}(G))}}_{\text{constant factor in } G} \,.$$

For fixed $w$ and bounded $\text{SHAP}(G)$, this scales as $c(G)\,\frac{\log N}{2}\dim(G)$ with $c(G) \in (0, 1]$. Hence the penalty still grows like $\log N$, under standard regularity conditions for BIC, this preserves large-sample consistency. Moreover, if $\text{SHAP}(G) = 0$ for all $G$ (e.g., no directional signal passes the confidence filter), XBIC reduces exactly to BIC.

**Hyperparameter $w$.** The weight $w$ trades off reliance on likelihood vs. directional evidence: larger $w$ lowers the relative penalty on edges supported by $\bar{\Phi}_{j \to i}$, typically increasing recall at some precision cost. We sweep $w$ and report the precision–recall trade-off in Section 4.3.

## 4 EVALUATION

We evaluate how XBIC improves discrete causal structure learning under varying data regimes. Experiments use samples generated from known Bayesian networks and span seven sample-size

settings to probe robustness. Because hill climbing with BIC is sensitive to data quantity Koller & Friedman (2009), we report results across all regimes.

## 4.1 SETUP

**Networks.** We use 10 benchmark discrete Bayesian networks from the `bnlearn` repository,[1] ranging from 6 to 76 variables and covering healthcare, insurance, weather, and software domains. Table 1 summarizes metadata.

Table 1: Summary of benchmark networks.

| Network | Nodes | Edges | Parameters | Avg. MB Size | Avg. Degree | Max In-degree | Domain |
|---------|-------|-------|------------|--------------|-------------|---------------|--------|
| Asia | 8 | 8 | 18 | 2.5 | 2.0 | 2 | Medical |
| Sachs | 11 | 17 | 178 | 3.09 | 3.09 | 3 | Cell Biology |
| Survey | 6 | 6 | 21 | 2.67 | 2.0 | 2 | Transport |
| Alarm | 37 | 46 | 509 | 3.51 | 2.49 | 4 | Medical |
| Child | 20 | 25 | 230 | 3.00 | 1.25 | 2 | Medical |
| Insurance | 27 | 52 | 984 | 5.19 | 3.85 | 3 | Insurance |
| Water | 32 | 66 | 10083 | 7.69 | 4.12 | 5 | Wastewater |
| Hailfinder | 56 | 66 | 2656 | 3.54 | 2.36 | 4 | Weather |
| Win95pts | 76 | 112 | 574 | 5.92 | 2.95 | 7 | Software |
| Hepar2 | 70 | 123 | 1453 | 4.51 | 3.51 | 6 | Medical |

**Baselines.** We compare to (i) hill climbing with standard BIC (BIC-HC), (ii) PC, and (iii) GES using the generalized score Huang et al. (2018). MMHC targets large sparse graphs and is not the focus here. For baselines that return a PDAG, we complete it to a DAG by randomly orienting undirected edges (while preserving acyclicity) before computing directed-edge metrics.

**Sample sizes.** For each network we generate seven data sizes, from $0.125\,M^2$ to $8\,M^2$ (where $M$ is the number of variables). Each setting is repeated 10 times; we report averages over identical splits across methods.

**Classifiers and SHAP.** For each target $X_i$ we train an XGBoost classifier (five-fold CV; Optuna search[2]) on $X_{\setminus i}$. Following Algorithm 1, TreeExplainer is applied only on instances whose predicted-class probability exceeds a fixed `CONFIDENCE_THRESHOLD`. Varying this threshold between 0.7 and 0.95 changed downstream $F_1$ by $< 1\%$ on average, suggesting the filter mainly reduces SHAP evaluations without materially affecting accuracy.

**Shapley weight.** We evaluate $w \in \{1, 2, 3\}$ in XBIC (note: $w = 0$ recovers BIC). Unless stated otherwise, XBIC refers to the indicated $w$.

## 4.2 METRICS

We report (i) precision: fraction of predicted *directed* edges that match the ground-truth direction; (ii) recall: fraction of ground-truth directed edges recovered; (iii) $F_1$: harmonic mean of precision and recall (emphasized due to sparsity); and (iv) structural Hamming distance (SHD): edge additions, deletions, and reversals to reach the ground truth (lower is better).

**Hyperparameter search.** For each target node we run Optuna (50 trials; 5-fold CV; RMSE objective) over the space in Table 3; the best configuration is refit on all data and its TreeSHAP values feed Stage 2.

## 4.3 RESULTS

Table 2 reports $F_1$ deltas of XBIC ($w$=2) versus baselines. On smaller networks and limited data, XBIC sometimes does not improve on BIC; this typically occurs when Stage 1 classifiers seldom surpass the confidence threshold, yielding few instances for reliable attributions and effectively reverting XBIC to BIC (SHAP($G$) near zero). For medium/large networks and moderate-to-large

---

[1] https://www.bnlearn.com/bnrepository/
[2] https://github.com/optuna

Table 2: F-score deltas of XBIC ($w{=}2$) relative to BIC, PC, and GES. Columns group sample sizes from $0.125\,M^2$ to $8\,M^2$. "–" marks GES runs that did not finish all of the runs.

| | Sample size | | | | | | | | | | | | | | | | | | | | |
| | $0.125M^2$ | | | $0.25M^2$ | | | $0.5M^2$ | | | $M^2$ | | | $2M^2$ | | | $4M^2$ | | | $8M^2$ | | |
| Network | BIC | PC | GES | BIC | PC | GES | BIC | PC | GES | BIC | PC | GES | BIC | PC | GES | BIC | PC | GES | BIC | PC | GES |
|---|---|---|---|---|---|---|---|---|---|---|---|---|---|---|---|---|---|---|---|---|---|
| Asia | 0 | 0 | 0 | 0 | 0 | 0 | 0.0 | **0.18** | 0.06 | 0.0 | **0.17** | -0.05 | **-0.12** | **0.22** | 0.04 | -0.01 | **0.18** | -0.05 | -0.02 | **0.16** | **0.07** |
| Sachs | 0 | 0 | 0 | 0.02 | **0.22** | **-0.08** | 0.06 | **0.28** | -0.07 | **0.15** | **0.21** | -0.01 | **0.2** | **0.25** | 0.02 | **0.18** | **0.2** | **0.03** | 0.08 | **0.07** | 0.01 |
| Survey | 0 | 0 | 0 | 0 | 0 | 0 | 0 | 0 | 0 | -0.09 | 0 | -0.02 | 0.07 | 0.1 | 0.04 | 0.05 | 0.07 | -0.01 | 0.09 | **0.14** | 0.2 |
| Alarm | **0.07** | **0.21** | **0.05** | **0.08** | **0.16** | -0.02 | **0.04** | **0.1** | - | **0.02** | **0.06** | - | **0.06** | 0.01 | - | **0.03** | 0.01 | - | **0.04** | **-0.05** | - |
| Child | 0.04 | **0.2** | **0.08** | 0.01 | **0.08** | 0.02 | -0.01 | **0.08** | 0.01 | 0.03 | **0.04** | 0.03 | **0.04** | **0.08** | 0.0 | 0.04 | **0.11** | - | -0.02 | **0.07** | - |
| Insurance | **0.08** | **0.18** | **0.05** | **0.11** | **0.18** | 0.01 | **0.09** | **0.14** | -0.03 | **0.1** | **0.19** | -0.03 | **0.08** | **0.17** | - | **0.07** | **0.21** | - | **0.09** | **0.14** | - |
| Water | -0.01 | **0.07** | **0.04** | 0.0 | **0.06** | **0.04** | 0.01 | **0.07** | **0.05** | 0.05 | **0.09** | - | 0.05 | **0.08** | - | 0.07 | **0.08** | - | 0.06 | **0.1** | - |
| Hailfinder | **0.08** | **0.13** | - | **0.08** | **0.18** | - | **0.12** | **0.12** | - | **0.1** | **0.18** | - | **0.13** | **0.14** | - | - | - | - | - | - | - |
| Win95pts | 0.0 | **0.11** | - | 0.0 | **0.06** | - | 0.02 | **0.04** | - | **0.07** | **0.05** | - | **0.07** | **0.04** | - | 0.0 | **-0.05** | - | -0.09 | **-0.15** | - |
| Hepar2 | 0.01 | **0.17** | - | 0.01 | **0.19** | - | 0.01 | **0.24** | - | 0.0 | **0.25** | - | 0.0 | **0.26** | - | -0.02 | **0.28** | - | 0.0 | **0.28** | - |

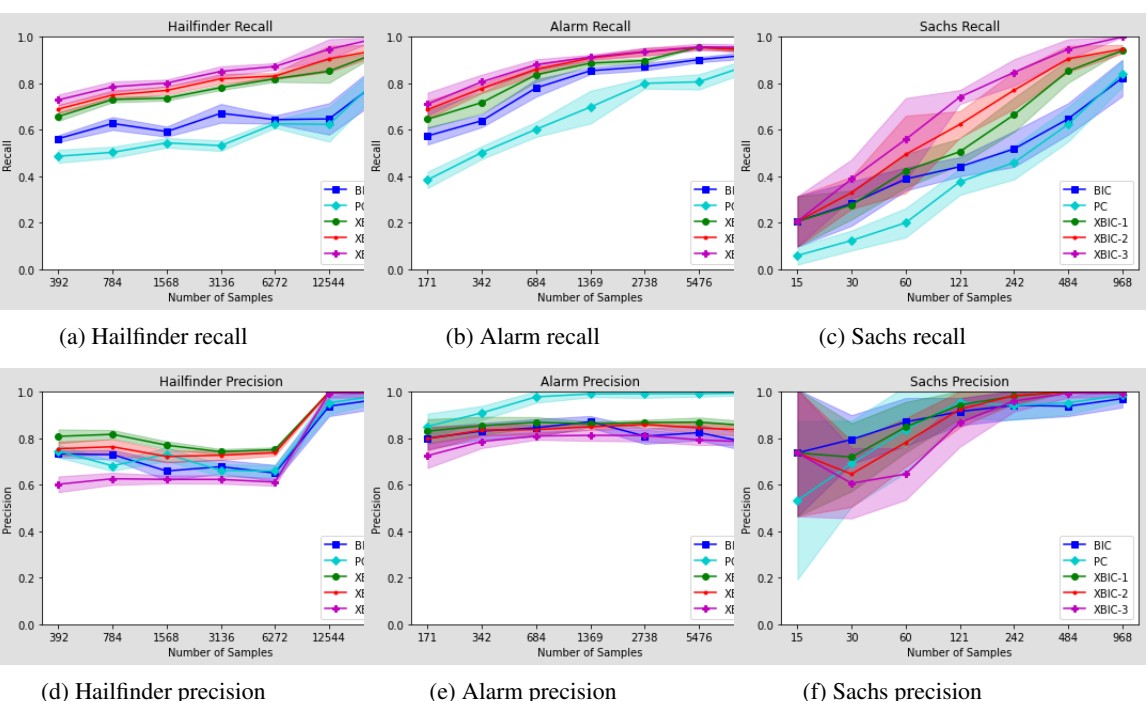

(a) Hailfinder recall      (b) Alarm recall      (c) Sachs recall

(d) Hailfinder precision      (e) Alarm precision      (f) Sachs precision

Figure 2: Precision and recall of XBIC with $w \in \{1, 2, 3\}$.

samples, XBIC generally improves $F_1$. We also examined precision and recall separately (Figure 2). Larger $w$ tends to increase recall (more edges admitted) while sometimes reducing precision, as expected from a softer penalty.

Table 4 aggregates improvements across all 700 runs. XBIC with $w{=}2$ attains the highest overall gain: $+5.6\%$ vs. BIC, $+20.9\%$ vs. PC, and $+9.6\%$ vs. GES (relative improvements).

We apply the adjusted Friedman test with $p < 0.05$ followed by Wilcoxon signed-rank tests. XBIC ($w{=}1$) and XBIC ($w{=}2$) significantly outperform all baselines; XBIC ($w{=}3$) significantly outperforms PC and is competitive with BIC, especially at smaller sample sizes where its higher recall is advantageous.

## 4.4 RUNTIME

All experiments ran on Linux with 4 CPUs and 20 GB RAM per task. Table 5 shows that XBIC is slower than BIC/PC due to classifier training and SHAP aggregation. GES exhibited poor scalability and often did not finish within 7 days on larger/denser networks.

Table 3: XGBoost hyperparameters (search space).

| Parameter | Range / Distribution | Notes |
|---|---|---|
| n_estimators | $\{50, 100, \ldots, 2000\}$ | step = 50 |
| eta | log-uniform $[0.01, 0.3]$ | learning rate |
| max_depth | int $[3, 20]$ | |
| subsample | uniform $[0.5, 1.0]$ | |
| colsample_bytree | uniform $[0.5, 1.0]$ | |
| $\lambda$ (L2) | log-uniform $[10^{-3}, 10]$ | |
| $\alpha$ (L1) | log-uniform $[10^{-3}, 10]$ | |
| min_child_weight | int $[1, 10]$ | |
| gamma | uniform $[0, 5]$ | min-loss reduction |

Table 4: Average $F_1$ improvement of XBIC over baselines across 700 runs.

| | Type | BIC | PC | GES |
|---|---|---|---|---|
| XBIC ($w$=1) | Relative | 5.1% | 20.2% | 9.0% |
| | Absolute | 0.03 | 0.11 | 0.05 |
| XBIC ($w$=2) | Relative | 5.6% | 20.9% | 9.6% |
| | Absolute | 0.04 | 0.12 | 0.06 |
| XBIC ($w$=3) | Relative | 2.5% | 17.3% | 6.3% |
| | Absolute | 0.02 | 0.10 | 0.04 |

Both the classifier training and TreeSHAP stages parallelize across targets (and, if desired, across instances), reducing wall-clock time in multi-core or distributed environments.

### 4.5 COMPARISON TO GES

GES exceeded the 7-day limit in many settings. For each (network, sample-size) pair, we retained only repetitions where GES completed and computed GES statistics on that subset. XBIC was compared head-to-head on the same repetitions. Despite this favorable filtering for GES, XBIC ($w$=1) and XBIC ($w$=2) achieved significantly lower SHD (paired $t$-test, $p < 0.05$), with improvements ranging from 6%–32% and 1%–27%, respectively; XBIC ($w$=3) showed no significant SHD advantage on this subset. Figure 3 visualizes the SHD differences.

### CONCLUSIONS

We presented XBIC, a score-based approach to causal structure discovery for purely discrete data that augments the Bayesian Information Criterion with edge-specific, directional Shapley evidence. By softly reducing the complexity penalty on edges with strong attribution support—while reverting to standard BIC when evidence is weak—XBIC guides hill-climbing toward orientations that are more consistent with the data-generating mechanisms.

On ten benchmark Bayesian networks (6-76 nodes) across seven sample-size regimes (700 runs), XBIC improved oriented-edge $F_1$ and reduced SHD relative to hill-climbing BIC, PC, and a generalized-score GES variant, with the strongest overall gains at $w$=2. These gains come with added computational cost from per-node classifiers and attribution aggregation. In our evaluation this cost was manageable for offline discovery, and both training and TreeSHAP computations parallelize naturally. XBIC sometimes offers little benefit on small samples, where the underlying classifiers seldom produce confident predictions and the method effectively defaults to BIC.

We see XBIC as especially relevant where variables are discrete, interventions are costly or infeasible, and interpretability matters (e.g., healthcare, environmental monitoring, risk and insurance).

Table 5: Average runtime per method (seconds).

| Network | BIC | PC | XBIC ($w=2$) |
|---------|-----|-----|------------|
| Asia | 0.39 | 0.09 | 74.78 |
| Sachs | 0.49 | 0.46 | 106.21 |
| Survey | 0.09 | 0.02 | 54.21 |
| Alarm | 9.30 | 12.22 | 523.52 |
| Child | 2.08 | 5.23 | 234.00 |
| Insurance | 4.51 | 10.78 | 382.15 |
| Water | 4.97 | 1.28 | 402.94 |
| Hailfinder | 36.47 | 15923.14 | 1904.25 |
| Win95pts | 75.33 | 33.91 | 2139.27 |
| Hepar2 | 40.33 | 130.98 | 1885.44 |

Figure 3: SHD difference (XBIC minus GES; lower is better). Negative values favor XBIC.

The method is a drop-in modification of a familiar score, facilitating adoption within existing BIC-based pipelines.

**Limitations and future work.** (i) *Runtime.* We will explore faster base learners (e.g., Light-GBM/CatBoost) and approximate or batched attribution to reduce overhead, alongside broader parallelization. (ii) *Small-sample regimes.* Adaptive confidence filtering and uncertainty-aware aggregation may strengthen directional signals when data are limited. (iii) *Search scalability.* Scaling beyond ∼75 nodes motivates hybrid or order-based searches (e.g., max–min hill climbing) and sparsity-aware neighborhoods. (iv) *Theory.* While XBIC preserves BIC in the absence of attribution signal, formal analysis of the weighting mechanism (e.g., bounds and convergence properties) is an important direction.

All code, datasets, and evaluation scripts are publicly available to foster reproducibility and further research.[3]

By bridging causal discovery and explainability, XBIC offers a practical route to more informative orientations in discrete causal graphs, complementing existing score-based methods while retaining their familiar workflow.

---

[3]https://anonymous.4open.science/r/causal_discovery_shap-6900/README.md

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
