# OpenReview forum: "XBIC: Shapley-Weighted BIC for Score-Based Causal Discovery in Discrete Bayesian Networks"
_ICLR.cc/2026/Conference — Submitted to ICLR 2026_

### Official Review · Reviewer_Evmc · 2025-10-28

**Soundness:** 1
**Presentation:** 1
**Contribution:** 1
**Rating:** 2
**Confidence:** 4

**Summary:**

The paper introduces XBIC, a modification of the Bayesian Information Criterion for discrete Bayesian network learning. XBIC adjusts BIC’s complexity penalty using edge-specific Shapley attribution values computed from per-node classifiers (using XGBoost). The idea is that when a candidate parent variable contributes strongly to predicting its child, the BIC penalty for including that edge is softened. The authors claim this helps orient edges within Markov equivalence classes. The method is evaluated on ten benchmark discrete networks (6–76 nodes) with synthetic data, compared to standard BIC hill-climbing, PC, and GES.

**Strengths:**

1.	Interesting empirical motivation.
The paper identifies a real problem: standard BIC scoring can sometimes underperform on discrete networks when local variable interactions are strong but sparsely represented. The attempt to incorporate model-based feature attribution (SHAP) into a scoring criterion is conceptually novel and could inspire follow-up research.
2.	Clear experimental setup.
The evaluation uses well-known benchmark networks (Asia, Child, Alarm, Hailfinder, etc.) from the bnlearn repository, which allows for some reproducibility and comparability of results.
3.	Potential for cross-pollination with machine learning methods.
Integrating XGBoost-based attribution measures into causal-structure scoring is a creative idea that could, with further development and theoretical grounding, suggest ways to combine predictive modeling and causal discovery frameworks.

**Weaknesses:**

1.	Unreliable benchmarking and unfair baselines.
The paper reports that GES “did not finish within seven days” on networks of 56–76 variables, which is inconsistent with known implementations (e.g., Tetrad, bnlearn) that complete such tasks in seconds. This strongly suggests an inefficient or incorrect implementation. As a result, the comparative runtime and performance claims are not credible.
2.	Lack of methodological clarity.
The XBIC algorithm is never presented in a self-contained form. There is no pseudocode, clear procedural description, or specification of how the SHAP-weighted penalty integrates into the search process. This makes it impossible to reproduce or verify the reported results.
3.	Weak theoretical grounding.
The proposed modification of BIC is heuristic. The claim that XBIC “preserves consistency” is asserted without proof, and no argument is given that the SHAP-weighted penalty yields correct model selection under standard assumptions. The theoretical contribution is therefore minimal.
4.	Excessive computational cost.
Each node-level model requires an XGBoost and SHAP computation, resulting in orders-of-magnitude higher runtime than standard BIC or GES. The paper acknowledges this overhead but does not justify it in terms of improved accuracy or scalability.
5.	Limited and potentially biased empirical evaluation.
Experiments are confined to small and medium discrete networks and omit widely used baselines such as MMHC, FGES with standard caching, and more recent hybrid or score-based methods. The evaluation also uses incomplete DAGs (from PC and GES) without orientation repair, which biases the results in favor of XBIC.
6.	Overstated claims.
The paper repeatedly asserts that XBIC “outperforms GES and PC” and that GES is impractical for moderately large networks. Given the evidence, these claims are not supported. The method is an exploratory heuristic, not a practical or theoretically justified replacement for established algorithms.
7.	Presentation issues.
The exposition is uneven: the technical content is embedded in prose, with critical steps scattered across sections. Key ideas (e.g., why SHAP values should adjust BIC penalties) are described only qualitatively, and notation is inconsistently introduced.

**Questions:**

1.	GES runtime discrepancy.
The paper reports that GES “did not finish within seven days” on networks with 56–76 variables. Could the authors clarify exactly which implementation was used (e.g., Tetrad, bnlearn, or a custom version), whether score caching was enabled, and if family scores were recomputed at every iteration? Since standard implementations complete these same benchmarks in seconds, such details are critical to understanding whether the reported runtime reflects algorithmic limitations or configuration choices.
2.	Algorithmic specification.
Algorithm 2 outlines a hill-climbing framework for structure learning with XBIC, but several key steps remain underspecified. Could the authors define precisely how the SHAP-weighted penalty modifies the BIC score, whether SHAP values are recomputed or reused after each graph modification, and how local score caching is handled? These aspects are necessary to reproduce the results and to evaluate the claimed runtime overhead.
3.	Fairness of baseline evaluation.
For PC and GES, the paper states that undirected edges were randomly oriented before computing F-scores. Random completions can introduce orientation noise and systematically penalize these methods. Would the authors consider evaluating on essential graphs (CPDAGs) or providing separate metrics for adjacency and orientation accuracy? Clarifying this would make the comparisons more interpretable.
4.	Theoretical justification.
Section 4.3 claims that XBIC “remains asymptotically consistent.” Could the authors specify under what assumptions this holds—e.g., whether the SHAP-weighted penalty satisfies score equivalence and decomposability—and provide at least a proof sketch or argument? Without these conditions, it is unclear how XBIC retains the statistical guarantees of standard BIC.
5.	Scalability claims.
The paper characterizes XBIC as “large-scale” and “dimension-adaptive,” yet all benchmarks involve fewer than 100 variables. Could the authors clarify whether “large-scale” refers to theoretical complexity, sample-size adaptation, or empirical scalability? If they expect the method to scale to hundreds or thousands of nodes, some runtime or complexity analysis would be helpful.
6.	Interpretability of SHAP weighting.
The proposed use of SHAP values as an adaptive regularizer is intriguing but not theoretically motivated. Could the authors provide intuition or formal reasoning showing why this weighting should improve orientation or structure accuracy beyond empirical observation? For example, is the weighting expected to approximate a sparsity-inducing prior or adjust for variable importance bias?
7.	Practical trade-offs.
Because each node-level model involves XGBoost training and SHAP evaluation, XBIC appears substantially more expensive than conventional score-based methods. Could the authors discuss concrete settings where this cost is justified—e.g., small, high-noise discrete datasets—or whether approximate SHAP computations or parallelization could mitigate runtime? Quantifying the trade-off would clarify XBIC’s practical niche.
8.	Additional baselines.
Since all datasets are discrete bnlearn networks, it would be informative to compare XBIC against modern discrete score-based algorithms such as BOSS (Andrews et al., 2023) or GRaSP, which achieve strong accuracy and scalability on the same benchmarks. Such comparisons would help determine whether XBIC’s reported improvements arise from the scoring function itself or from implementation differences in the search procedure.

---

### Official Review · Reviewer_bpNA · 2025-10-30

**Soundness:** 2
**Presentation:** 2
**Contribution:** 2
**Rating:** 2
**Confidence:** 3

**Summary:**

The paper proposes XBIC, a BIC-style score that down-weights the model complexity penalty using edge-level “directional evidence” derived from TreeSHAP attributions of per-node classifiers. The search remains standard hill-climbing, but the BIC penalty is scaled by (Eqs. 2–3). Experiments on 10 discrete BN benchmarks and seven sample-size regimes report average oriented-edge F1 gains over BIC/PC/GES.

**Strengths:**

1. The paper clear frames the Markov-equivalence orientation problem and a “drop-in” modification to a familiar score.
2. The paper proposes a Transparent algorithmic pipeline aiming to solve the problem (per-node XGBoost → TreeSHAP aggregation → hill-climbing with XBIC).
3. The paper's algorithm shows average improvements (e.g., +5.6% F1 vs. BIC overall for w=2; +20.9% vs. PC) compared to traditional methods.

**Weaknesses:**

1. Runtime overhead is severe. Runtimes are orders of magnitude larger than BIC/PC (e.g., Asia: 0.39s BIC vs. 74.78s XBIC; Win95pts: 75.33s vs. 2139.27s), which materially limits practical use.

2. All data are generated from canonical bnlearn BNs; there are no real-world observational datasets with accepted ground-truth DAGs or interventional verification. Orientation is assessed mainly via directed-edge F1 after forcing DAGs, rather than CPDAG-aware metrics.

3. The headline improvements average absolute +0.04 F1 vs. BIC (w=2). Per-dataset tables show several regimes with negligible or negative deltas and sensitivity to w. The paper itself notes small-sample regimes where XBIC effectively reduces to BIC (few confident predictions).

4. Evaluation design likely favors the proposed method. PDAG completion by random orientation: For baselines that output CPDAGs (e.g., PC), the paper randomly orients undirected edges to form a DAG before directed-edge scoring. This can artificially depress baseline precision/recall on directed edges. More appropriate is CPDAG-aware metrics or reporting “correctly oriented compelled edges.” XBIC enjoys heavy per-node tuning (Optuna over many XGBoost hyperparameters) while there is no evidence that comparable hyperparameter sweeps were performed for the baselines (e.g., PC test selection, GES score variants).

5. The complexity term is multiplied by exp(w·SHAP(G)). Beyond a brief “bounded SHAP” remark, no conditions are proven under which SHAP(G) remains bounded or leads to asymptotically correct orientations; the consistency discussion is qualitative and hinges on assumptions not verified empirically.

**Questions:**

1. Can you add real-data case studies (with partial ground truth or interventional checks) and an ablation on the base learner (e.g., LightGBM/CatBoost) and the confidence threshold 𝜏?
2. Can you provide CPDAG-level metrics (e.g., on compelled edges) to fairly assess PC/constraint methods without random DAG completion?
3. Please report validation protocol for selecting w (global constant vs. per-dataset tuning).
4. How sensitive are results to the absolute-value choice in Eq. (3)? What happens if signs are retained or if you contrast Φ̄_{j→i} vs. Φ̄_{i→j}?

---

### Official Review · Reviewer_Z5SS · 2025-10-31

**Soundness:** 2
**Presentation:** 2
**Contribution:** 2
**Rating:** 2
**Confidence:** 2

**Summary:**

The paper presents a novel  score based causal discovery approach for purely discrete data. The paper compares against other score based methods and reports improvements in relatively small datasets.

**Strengths:**

The paper is nice and easy to read, the visualizations are pleasant to the eye and it is clearly communicated what is done and what the approach can do.

**Weaknesses:**

Interestingly enough the provided link leads to an anonymous repo which is then empty. Not sure if that is on purpose.

In recent years amortized approaches similar to prior fitted networks have been very popular and are probably SOTA approaches for causal structure learning [1,2] or cause effect estimation [3] with many concurrent works at the same time (i just provide pointers from which it would be easy to get find the others). It is unclear to me why the amortized causal discovery methods should not be a valid baseline here and why they should not beat the proposed approach by a wide margin in the empirical studies.

Beyond that the comparisons with GES or PC are basically very old baselines and seem very selective.

[1] Ke, Nan Rosemary, et al. "Learning to induce causal structure." arXiv preprint arXiv:2204.04875 (2022).
[2] Lorch, Lars, et al. "Amortized inference for causal structure learning." Advances in Neural Information Processing Systems 35 (2022): 13104-13118.
[3] Ma, Yuchen, et al. "Foundation Models for Causal Inference via Prior-Data Fitted Networks." arXiv preprint arXiv:2506.10914 (2025).

**Questions:**

See weaknesses.

---

### Official Review · Reviewer_DJmD · 2025-10-31

**Soundness:** 3
**Presentation:** 2
**Contribution:** 2
**Rating:** 2
**Confidence:** 4

**Summary:**

This paper introduces XBIC, a method to learn the structure of causal graphs. XBIC uses a classifier to predict the value of nodes given the other nodes, computes a Shapley value, and then augments the traditional BIC score with the Shapley value to more accurately compute causal graphs using a hill-climbing/iterative method for constructing the graph.
Empirical results are provided on small discrete Bayesian networks showing that XBIC generally achieves better results than the competing methods.

**Strengths:**

+ The proposed method is relatively clearly described and well motivated.
+ The empirical results seem to demonstrate the superiority of XBIC compared to PC and GES.
+ The available code and results are a good contribution to open science.

**Weaknesses:**

+ The comparisons in Section 4.3 are all done with respect to PC and GES, which are relatively old algorithms in the field of causal discovery. I am not very familiar with this discrete setting, so I can't point to algorithms that should be compared to in 2025. However it seems there should be others that it could be compared to, such as an MCMC-based approach.
+ The presentation is not very polished. In particular, the graphs in Figure 2 have their label cut off in the first two columns. Much of the space is taken up by large graphs that could be made smaller.
+ It is not clear to me whether the significance of this work is sufficient and appropriate to be accepted at ICLR. In terms of the novelty, the application is relatively limited to discrete small graphs, the actual application utility of these graphs is relatively limited in practical terms. Secondly, the venue fit is very good. ICLR is specifically a venue about deep learning and representation learning, which this work does neither of. The work uses a more classical iterative method, which doesn't use any representation learning. I think a venue such as AISTATS or KDD might be more appropriate for this work.

**Questions:**

+ Are there any more recent algorithms that the proposed approach can be compared to?
+ Can the authors elaborate on their contributions and why they are a good fit for the conference?

---

### Meta-Review · Area_Chair_HhR9 · 2026-01-13

**Summary:**

This submission introduces a weighted BIC score, where the weights are induced by Shapley values. Empirical evaluation shows improvements. Reviewers were unanimously negative: Common concerns include limited and inconsistent evaluation, clarity issues, lack of theory, feasibility, and overclaiming.

**Reviewer Concerns:**

Authors did not respond.

**Reviewer Scores:**

No changes.

---

### Decision · Program_Chairs · 2026-01-26

Reject